# Separation of Coiled-Coil Structures in Lamin A/C Is Required for the Elongation of the Filament

**DOI:** 10.3390/cells10010055

**Published:** 2020-12-31

**Authors:** Jinsook Ahn, Soyeon Jeong, So-Mi Kang, Inseong Jo, Bum-Joon Park, Nam-Chul Ha

**Affiliations:** 1Department of Agricultural Biotechnology, Centre for Food and Bioconvergence, and Research Institute for Agriculture and Life Sciences, CALS, Seoul National University, Seoul 08826, Korea; jinsook1113@snu.ac.kr (J.A.); syjeong0711@snu.ac.kr (S.J.); inseong89@kobiolabs.com (I.J.); 2Department of Molecular Biology, College of Natural Science, Pusan National University, Busan 46241, Korea; rosa.somi.kang@hanmail.net (S.-M.K.); bjpark1219@pusan.ac.kr (B.-J.P.)

**Keywords:** nuclear lamin A/C, filament assembly, EDMD, assembly mechanism, laminopathies, eA22 interaction, intermediate filament, structural transition, molecular regulator

## Abstract

Intermediate filaments (IFs) commonly have structural elements of a central α-helical coiled-coil domain consisting of coil 1a, coil 1b, coil 2, and their flanking linkers. Recently, the crystal structure of a long lamin A/C fragment was determined and showed detailed features of a tetrameric unit. The structure further suggested a new binding mode between tetramers, designated eA22, where a parallel overlap of coil 1a and coil 2 is the critical interaction. This study investigated the biochemical effects of genetic mutations causing human diseases, focusing on the eA22 interaction. The mutant proteins exhibited either weakened or augmented interactions between coil 1a and coil 2. The ensuing biochemical results indicated that the interaction requires the separation of the coiled-coils in the N-terminal of coil 1a and the C-terminal of coil 2, coupled with the structural transition in the central α-helical rod domain. This study provides insight into the role of coil 1a as a molecular regulator in the elongation of IF proteins.

## 1. Introduction

Intermediate filament (IF) proteins provide vital mechanical support in higher eukaryotic cells, with various physical properties when they form filamentous polymers [1,2,3]. The monomeric units of all IF proteins share a tripartite structural organization: a central α-helical rod domain flanked by non-α helical N-terminal head and C-terminal tail regions [4,5,6,7]. The α-helical rod domain mostly consists of multiple heptads and hendecads, imparting a high propensity to form parallel coiled-coil dimers with different superhelical pitches. The α-helical rod domain is further divided into three segments (coil 1a, coil 1b, and coil 2) by flanking linkers (L1 and L12, respectively) lacking periodic repetitions [7]. The stutter region within coil 2 shows a different periodic rule from the adjacent areas [8,9]. Two IF consensus sequence motifs are found at both end regions of the central rod domain (Figure 1a), which are highly conserved among IF proteins and play a crucial role in forming the filamentous structure [10,11,12,13,14]. The α-helical rod domains are assembled to produce a long filamentous structure with 10 nm thickness in cytoplasmic IF proteins [15,16,17].

Nuclear IF lamin A/C is essential in forming and maintaining the nuclear structure by providing rigidity and flexibility underneath the inner nuclear membranes [21,22,23]. The parallel coiled-coil dimers of lamin A/C, as a fundamental building block, assemble laterally and longitudinally into high-order filamentous structures [23,24]. Recent in situ cryoelectron tomography (cryo-ET) images visualized the 3.5 nm thick filament of lamin A/C, decorated by an Ig-like fold domain at both sides with 20 nm longitudinal intervals, unlike 10 nm thick cytosolic IF proteins [17]. The assembly model of lamin A/C was substantially detailed at a higher resolution by the crystal structure of the N-terminal half fragment of lamin A/C, called the lamin 300 fragment. The crystal structure presented the structural features of the so-called “A11 tetramer,” which consists of two coiled-coil dimers in an antiparallel manner by overlapping the coil 1b regions [18,25]. The crystal structure and biochemical analyses indicate “eA22 binding mode” for joining the A11 tetramers. The parallel overlapping between the C-terminal region of coil 2 and the coil 1a region in the A11 tetramer was crucial for the eA22 interaction to form the eA22 binding mode. However, the overlapping length between the A11 tetramers in the eA22 binding mode is under debate [18,26,27]. Thus, further study is required to understand the molecular mechanism for the elongation of IF proteins.

Many genetic mutations in the lamin A/C gene result in human diseases, such as Emery Dreifuss muscular dystrophy (EDMD), cardiomyopathy dilated (CDM1A), Dunnigan-type familial partial lipodystrophy (FPLD), and Hutchinson-Gilford progeria syndrome (HPGS) [28,29,30,31,32]. The genetic variations causing diseases were highly associated with two IF consensus motifs interfacing parallel overlapping regions between coil 1a and coil 2 [10,12,13,14,18]. In this study, we first investigate the role of the eA22 interaction in lamin-related diseases. We then propose a binding model for the eA22 interaction remodeling the coiled-coil interactions, coupled with the structural transition in the central α-helical rod domain.

## 2. Materials and Methods

### 2.1. Plasmid Construction

The genes encoding each protein were amplified DNA fragments coding for residues 1-125 (L35V, L38C, L42C, Y45C, L59R, and I63S mutants), 1-300 (L35V and I63S mutants), and 250-400 (wild type, Y267C, K316C, L362C, L380C, L380A mutants) of lamin A/C and were inserted into the pProEx-HTa vector for overproduction of the lamin fragments (Thermo Fisher Scientific, Waltham, MA, USA). The used oligonucleotide primer sequences are listed in Appendix A. The resulting plasmids encode the N-terminal His-tag and the tobacco etch virus (TEV) protease cleavage site at the N-terminus of the lamin proteins. Plasmids of lamin 1-125 (wild type), 1-300 (wild type, Y45C, and L59R), and 250-400 were obtained, as used in previous research [18]. The plasmids were transformed into the *E. coli* BL21 (DE3) strain for overexpression. For immunofluorescence staining, the amplified DNA fragments encoding wild-type and the L35V, Y45C, L59R, and I63S mutants of full-length lamin A/C were inserted into the pcDNA3.1(+) vector (Thermo Fisher Scientific, Waltham, MA, USA).

### 2.2. Purification of Recombinant Proteins

The transformed *E. coli* cells were cultured in 3 L of Terrific Broth medium containing 100 μg/mL ampicillin and 34 μg/mL chloramphenicol at 37 °C to an OD600 of 1.0. The expression of proteins was induced using 0.5 mM IPTG at 30 °C for 6 h. After cell harvest by centrifugation, cells were resuspended in lysis buffer containing 20 mM Tris-HCl (pH 8.0), 150 mM NaCl, and 2 mM 2-mercaptoethanol. The cells were disrupted using a continuous-type French press (Constant Systems Limited, Daventry, UK) at 23 kpsi pressure. The cell debris was removed after centrifugation at 19,000 g for 30 min at 4 °C. The supernatant was loaded onto a cobalt-Talon affinity agarose resin (GE Healthcare, Menlo Park, California, USA). The target protein was eluted with lysis buffer supplemented with 250 mM imidazole and 0.5 mM EDTA. The lamin 250-400 protein was treated with TEV protease to cleave the His-tag. Then the target proteins were loaded onto a HiTrap Q column (GE Healthcare, Menlo Park, California, USA). An increasing linear gradient of NaCl concentration was applied onto the HiTrap Q column for further purification. The fractions containing the protein were pooled and applied onto a size-exclusion chromatography column (HiLoad 26/600 Superdex 200 pg, GE Healthcare, Menlo Park, California, USA) pre-equilibrated with lysis buffer. The purified protein was concentrated and stored at −80 °C until used for biochemical assays.

### 2.3. Pull-Down Assays

A pull-down assay was conducted using His-tagged lamin proteins immobilized on the Ni-NTA resin as bait. His-tag cleaved lamin proteins as prey were incubated on the His-tagged lamin immobilized resin pre-equilibrated in a 20 mM Tris-HCl (pH 8.0) buffer containing 150 mM NaCl (or 50 mM NaCl) at room temperature for 30 min. After washing with the lysis buffer supplemented with 20 mM imidazole, the remaining resin and the fractions were analyzed using SDS-PAGE. The experiments for pull-down assay were performed under either the oxidized condition (0.5 mM GSSG) or the reducing condition (5 mM tris(2-carboxyethyl) phosphine; pH 8.0).

### 2.4. Isothermal Titration Calorimetry (ITC)

ITC experiments were conducted using an Auto-iTC200 Microcalorimeter (GE Healthcare) at the Korea Basic Science Institute (Ochang, Korea). His-tagged proteins of wild-type and L35V, Y45C, L59R, or I63S mutants of lamin 300 fragments (25 μM; 0.7 mg/mL) were prepared in the sample cell (370 μL), and the TEV protease-cleaved lamin 250-400 fragment (180 μM; 3 mg/mL) was loaded into the injectable syringe. All samples were dialyzed against PBS overnight before the ITC experiments. Titration measurements of 19 injections (2 μL) with 150 s spacing were performed at 25 °C while the syringe was stirred at 750 rpm. The data were analyzed using MicroCal OriginTM software.

### 2.5. Circular Dichroism

Circular dichroism (CD) spectra were collected from a Chirascan plus CD spectrometer (Applied Photophysics, Surrey, UK). His-tagged lamin 1-125 fragments of wild-type and L35V, Y45C, L59R, or I63S mutants in PBS (1 mg/mL) were subjected to CD experiments.

### 2.6. Immunofluorescence Staining

A human fibrosarcoma cell line (HT1080) obtained from ATCC was maintained in liquid Dulbecco Modified Eagle Medium (DMEM0 containing 10% (*v/v*) FBS and 1% (*v/v*) antibiotics at 37 °C. HT1080 cells were seeded on glass coverslips and transfected with the vectors expressing wild-type and mutants of full-length lamin A using jetPEI (Polyplus Transfection). Cells were rinsed briefly in PBS after 24 h transfection. After fixing with 99% methanol (stored at −20 °C for at least 2 h before use) for 15 min at −20 °C, cells were permeabilized with 0.1% (*v/v*) Triton X-100 for 5 min and incubated with blocking buffer (5% normal goat serum (31873; Invitrogen) in PBS) for 1 h at room temperature. After washing briefly with PBS, the cells were incubated with an anti-lamin A/C (sc-376248; Santa Cruz Biotechnology) primary antibody (1:200; diluted in blocking buffer) in blocking buffer overnight at 4 °C, followed by a secondary antibody (anti-mouse Ab-FITC; 1:400; diluted in blocking buffer) for 7 h at room temperature in the dark. The nuclei of cells were stained with DAPI for 10 min. After washing 3 times quickly in PBS, the coverslips were applied with an antifade mounting medium. The signals of immunofluorescence were detected by fluorescence microscopy (Logos).

## 3. Results

### 3.1. Mutations of Laminopathies Altered the eA22 Interaction

In a previous study, we investigated the Y45C and L59R mutations in coil 1a in terms of the eA22 interaction [18]. Mutation of Y45C in EDMD (137A>G) abolished the eA22 interaction [33], while the L59R mutation in CDM1A (197T>G) increased the eA22 interaction [34]. These observations suggest that lamin-related diseases (laminopathies) such as EDMD and CDM1A are associated with increased and decreased eA22 interaction compared to wild-type lamin A/C. To extend the relationship between the mutations and the strength of the eA22 interaction, we investigated two additional mutations, L35V and I63S in EDMD (103C>G, 188T>G) [15,33,35,36]. All four mutations (L35V, Y45C, L59R, and I63S) were in the *a* or *d* position of coil 1a in the crystal structure [18], which potentially affects the coiled-coil interaction in the coiled-coil dimeric unit (Figure 1) [29,31,37,38,39].

We purified four mutant proteins based on the lamin 300 fragment (residues 1-300) as the coil 1a-containing proteins to examine the eA22 interaction. The mutant proteins were expressed as a dimeric unit in solution, like the wild-type fragment. As previously described, we performed the His-tag pull-down assay to evaluate the strength of the eA22 interaction [18]. The His-tagged mutant lamin 300 fragments were incubated with the C-terminal part of coil 2 (residues 250-400) in a low-salt buffer containing 50 mM NaCl or a high-salt buffer containing 150 mM NaCl. The Y45C mutation reduced the binding strength of the eA22 interaction in both buffers (Figure 2a). In contrast, L59R and I63S increased the eA22 interactions in both buffer conditions (Figure 2a). The L35V mutation did not affect the binding strength of the eA22 interaction, based on the experiment using the lamin 300 fragment. However, the L35V mutation in the lamin 1-125 fragment showed a substantial reduction in the eA22 interaction (Figure 2). Thus, our results potentiated the hypothesis that the higher or lower strength of the eA22 interaction is closely related to the laminopathies.

We next employed isothermal titration calorimetry (ITC) to analyze the binding affinities quantitatively. The L59R mutation increased the eA22 interaction by ~56 fold according to the K_D_ values measured in this study, consistent with the results reported in the previous study (Figure 3a,d) [18]. The I63S mutation increased the eA22 interaction by ~2 fold (Figure 3e). In contrast, the L35V and Y45C mutations decreased the eA22 interaction by ~2 fold and ~5 fold, respectively (Figure 3b,c). These results are in good agreement with the results from the pull-down assays shown in Figure 2.

To examine the effects of the mutations on the nuclear shape and distribution of lamin A/C in cells, we overexpressed the mutant lamin genes in the HT1080 cell lines. Lamin A L35V and Y45C, with a lower eA22 interaction, were localized mainly throughout the nucleoplasm and formed blebs near the nuclear envelope in the cytoplasm (Figure 4c,d). This phenomenon appears to result from the fact that the mutant lamin molecules with the lower eA22 interaction could not be correctly incorporated into the nuclear lamina in the nucleoplasm. However, lamin A L59R and I63S, with a higher eA22 interaction, presented strong lamin aggregates at the peripheral region of the nuclear envelopes in the cytoplasm (Figure 4e,f). These observations can be interpreted that the mutant lamins assemble already in the cytoplasm by the higher tendency of the oligomerization, which prevented the import of the lamin proteins into the nucleus and the incorporation into the lamina. Alternatively, since the mutant lamins cannot be completely disassembled when the nucleus breaks down at mitosis and during nuclear assembly, they could not be incorporated in the nuclear lamina and stay filamentous structures in the cytoplasm. The results from Y45C and L59R were similar to previous observations [18]. Our findings confirm the previous proposition that both stronger and weaker eA22 interactions result in adverse effects on the formation of robust nuclear structures than the proper eA22 interaction of wild-type lamin.

### 3.2. The eA22 Interaction Requires Separation of the Coiled-Coil Dimer in the Coil 1a Region

To explain the mutations affecting the eA22 interaction from a structural view, we closely examine the mutations in the dimeric structure of the lamin 300 fragment (Appendix A) [18]. The modeled structure of the L35V mutant suggests that the mutation would stabilize the coiled-coil dimer at the coil 1a region because the distance between coiled-coils would decrease. According to the ΔΔGu (Ala) values indicating the stability of the coiled-coil structure at the a and d positions [40,41], valine stabilizes the coiled-coil structure of coil 1a compared to the leucine residue (Appendix A).

Tyr45 is conserved among vimentin as well as lamin A and B families. Tyrosine residues at the a and d positions destabilize the coiled-coil structure in general [18,41]. Thus, the Y45C mutation was suggested to stabilize the coiled-coil interaction of coil 1a because the cysteine residue is a better fit at the d position for coiled-coil formation (Appendix A) [18]. Conversely, loss of their hydrophobicity in the a or d heptad position seems to be why the mutations of L59R and I63S weakened the coiled-coil dimer structure. In particular, the modeled structure of the L59R mutant showed that Arg59 sterically clashed with and repulsed each other due to their bulky and positive charge (Appendix A) [18].

We explored the structural features of mutant lamin proteins focusing on the coil 1a region using circular dichroism (CD) spectroscopy. The CD spectra demonstrate the propensity for the secondary structure of the proteins. Since the separated α-helices in solution would not be as stable as the coiled-coil dimers, the α-helical tendency of the coil 1a region is a good indicator of the stability of the coiled-coil dimers [40]. To exclude the background α-helicity in the resting region, we used shorter fragments covering residues 1-125 containing the N-terminal head, coil 1a, L1, and a small portion of coil 1b instead of the lamin 300 fragment. The shorter fragments harboring the mutations exhibited similar results to those with the longer lamin 300 fragments in terms of an alteration of the eA22 interaction (Figure 2b and Appendix A). The CD spectra of the shorter fragments showed that the Y45C mutant protein has the highest α-helical content of 54%, indicating the largest increase in stability of coiled-coil dimers (Figure 5b; red). The L59R mutant has the lowest α-helical content of 16%, presumably indicating substantial disruption of coiled-coil dimers (Figure 5b; green). The L35V and I63S proteins showed slight differences in terms of α-helical content, which agree well with the relatively mild affinity changes to the coil 2 region seen in ITC (Figure 5b; yellow and purple). Thus, our findings suggest that the degree of α-helix tendency might be reversely correlated to the strength of the eA22 interactions, presumably because the coiled-coil interaction would stabilize the α-helical tendency.

### 3.3. Leu38 and Leu42 at the N-Terminal End Region of Coil 1a Are Vital in the Interaction with the Coil 2 Region

The L38C and L42C mutant proteins in the shorter N-terminal fragment (residues 1-125) were generated to analyze the complete prevention of separation of the coiled-coils in coil 1a on the formation of the eA22 interaction. Since Leu38 and Leu42 residues are at the a and d positions and in close contact in the coiled-coil structure, changing these residues to cysteine would form a disulfide bond under oxidized conditions, resulting in the prevention of separation of the coil-coil structure (Figure 1). Unexpectedly, mutant proteins did not bind to the coil 2 fragment under either reduced or oxidized conditions (Appendix A). These findings indicate that Leu38 and Leu42 are essential for the eA22 interaction, presumably to bind with the coil 2 region. 

### 3.4. The eA22 Interaction Requires Separation of Coiled-Coil Dimers in the C-Terminal Region of Coil 2

Using the same strategy as in the coil 1a part, we explored the role of the coil 2 part in the eA22 interaction. The C-terminal region of coil 2 can be divided into two subparts by the stutter region (residues 320-330; Figure 1a). The stutter region is essential for the elongation of the lamin and the other cytosolic IF proteins [7,42]. The crystal structures of these regions of lamin B1 and vimentin represented that the C-terminal region of coil 2 containing the stutter region maintained the coiled-coil α-helical conformation with the corresponding coiled-coil patterns [8,9,43].

To test if the separation of the coiled-coils is required during the eA22 interaction, we introduced four single mutations at the a or d positions of the heptad or hendecad pattern in the coil 2 fragment. Each residue was replaced with cysteine to form a disulfide bond in the oxidized condition, resulting in blocked separation of the coiled-coil dimer. Two residues were selected before the stutter region and the others after the stutter region. Of four mutants (Y267C, K316C, L362C, and L380C), the L380C mutation abolished the eA22 interaction with the coil 1a fragment even in the reduced condition, indicating that Leu380 is essential in the interaction with the coil 1a part (Figure 6a and Appendix A). The subsequent L380A mutant protein confirmed the importance of Leu380 in the eA22 interaction (Figure 6b and Appendix A).

Importantly, the L362C mutation, located after the stutter region, abolished the eA22 interaction only under the oxidized condition, allowing the disulfide bond that prevents rearrangement of the coiled-coil structure of the C-terminal region of coil 2 (Figure 6b and Appendix A). In contrast, the other two mutations (Y267C and K316C), located before the stutter region, did not affect the eA22 interaction under either reduced or oxidized conditions (Figure 6a and Appendix A). These findings indicate that separation of the coiled-coils at the C-terminal region of coil 2 after the stutter region is required for the eA22 interaction.

## 4. Discussion

In this study, we observed the altered binding affinity of the eA22 interaction by introducing the four mutations related to laminopathies into coil 1a. Based on the structural and biochemical analysis, as a result, the eA22 interaction was weakened in the mutations, which induced more stable coiled-coil dimers in coil 1a. In contrast, the mutations that cause the instability of coiled-coil (CC) dimers due to the loss of hydrophobic properties enhanced the eA22 interaction more than the wild type. Furthermore, we showed that the eA22 interaction also requires the CC-separation of the C-terminal part of coil 2. These results imply that the dynamic conformational changes in the N-terminal of coil 1a and C-terminal of coil 2 are necessary for maintaining the normal function of nuclear lamin.

To account for the conformational transition of coiled-coil dimers both in the N-terminal of coil 1a and the C-terminal of coil 2 parts in the eA22 interaction, we closely examined the structures of the lamin 300 fragments and vimentin (residues 99–189), both of which contain coil 1a, coil 1b, and flanking linker L1 (Figure 1b and Appendix A) [18,40]. Both structures exhibited all α-helical conformation in the coil 1a and coil 1b regions, including the flanking linker that did not follow the heptad rule. However, the two structures presented quite a different arrangement in terms of coiled-coil dimer formation and bending of α-helices. We noted that the abrupt bending was found at the linker L1 region connecting coil 1a to coil 1b in the continuous coiled-coil structure and that linker L1 does not follow the heptad rule (Figure 1b). In contrast, no abrupt kink was found in the vimentin structure with separation of the coiled-coil in the coil 1a region, also caused by the broken heptad rule at the linker (Appendix A). The structural comparison suggests that the kink at the linker compensates for the broken heptad rule, resulting in the formation of stable coiled-coil interaction throughout the coil 1a and coil 1b regions. Otherwise, the coiled-coil would be separated without the kink at the linker region. Thus, our findings suggest a dynamic equilibrium between two conformations: kinked and CC separated conformation.

We next applied a similar mechanism in the C-terminal region of coil 2. We observed that blocking the coiled-coil separation after the stutter region abolished the eA22 interaction (Figure 6). Significant conformational rearrangement, accompanied by coiled-coil separation, would occur at the C-terminal part of coil 2 for the eA22 interaction. The stutter region does not follow the heptad rule, as in linker L1, and the rest of the coil 2 region loosely follows the heptad rule. We further expect that the four-helix bundle model, previously proposed by Herrmann and Strelkov et al. [11,44,45], represents the resulting complex structure of coil 1a and coil 2.

Based on the dynamic conformational equilibrium, we propose an assembly model for the eA22 interaction in the elongation process of IFs (Figure 7). Until the cognate partner proteins appear, the A11 tetramer formed transiently in lamin A/C has the dynamic equilibrium between two conformations; kinked and coiled-coil (CC) separated conformation. Two A11 tetramers then produce the eA22 interaction by the straight and separated conformation for the elongation of the filament. This assembly model has many advantages in accounting for selective assembly of the IF members to identify the appropriate binding partners among more than 67 members of IF proteins with a similar structural organization [46,47,48]. In the mismatched association, the complex would be dissociated into the kinked coiled-coil structures.

Dysfunctions of coil 1a caused by genetic mutation would deteriorate dynamic remodeling or correct mesh formation in cells for the flexible and robust nuclear envelope, resulting in laminopathies such as EDMD and DCM. In this study, our findings imply that the dynamic conformational changes of coil 1a and coil 2 are necessary for maintaining the normal function of nuclear lamin. Although more biochemical and genetic studies are required to demonstrate the assembly mechanism, our study provides a detailed molecular view of nuclear lamins and IF proteins’ structures in higher eukaryotes. This study will improve our understanding of the various biological processes and diseases related to IF proteins since both the weakened and augmented eA22 interactions are related to lamin-related disorders.

## Figures and Tables

**Figure 1 cells-10-00055-f001:**
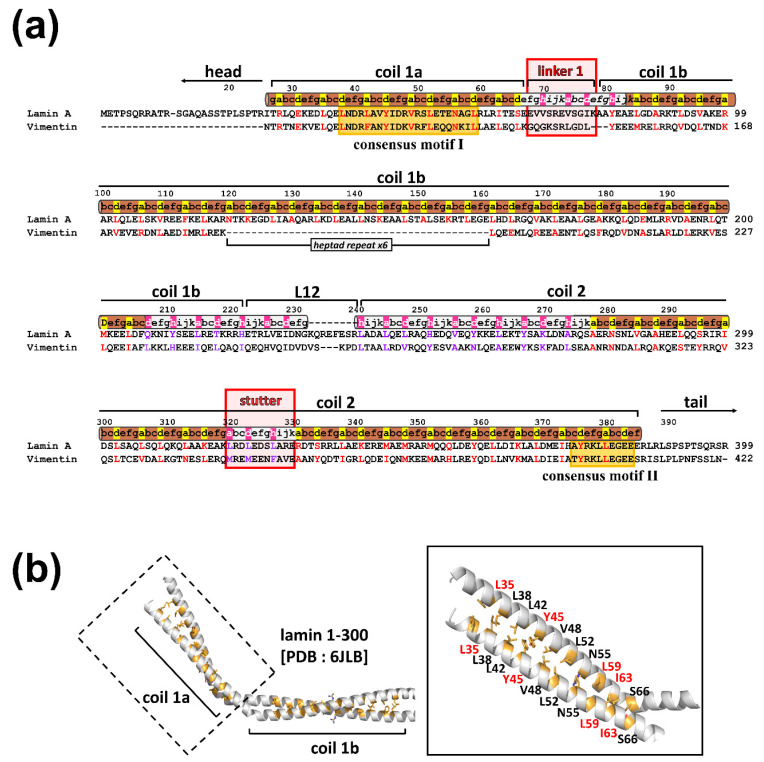
Kinked coiled-coil structure of the N-terminal region of coil 1a and coil 1b parts in lamin A/C (PDB 6JLB [18]). (**a**) Sequence alignment of the central rod domain in lamin A/C and vimentin proteins. Pairwise sequence alignment of lamin (human lamin A/C; P02545.1) and putative intermediate filament (IF) proteins (human vimentin; NP_003371.2). Sequence periodicity in the heptad (brown) or hendecad (gray) repeat, which was previously predicted [19,20] and revealed by the crystal structure of lamin 1-300 [18], is displayed above the sequences. The yellow boxes indicate highly conserved residues (consensus motifs I and II). Linker 1 and stutter regions are highlighted in red boxes. The *a*, *d*, and *h* residues are highlighted in red for heptad and purple for hendecad repeats. (**b**) The ribbon diagram presents an N-terminal area composed of coil 1a, linker 1, and half of coil 1b. The coil 1a region is enlarged in the box (right). The interhelical hydrophobic residues at the a and d positions of the heptad repeat are shown as yellow sticks. The residues for further mutational studies are labeled in red.

**Figure 2 cells-10-00055-f002:**
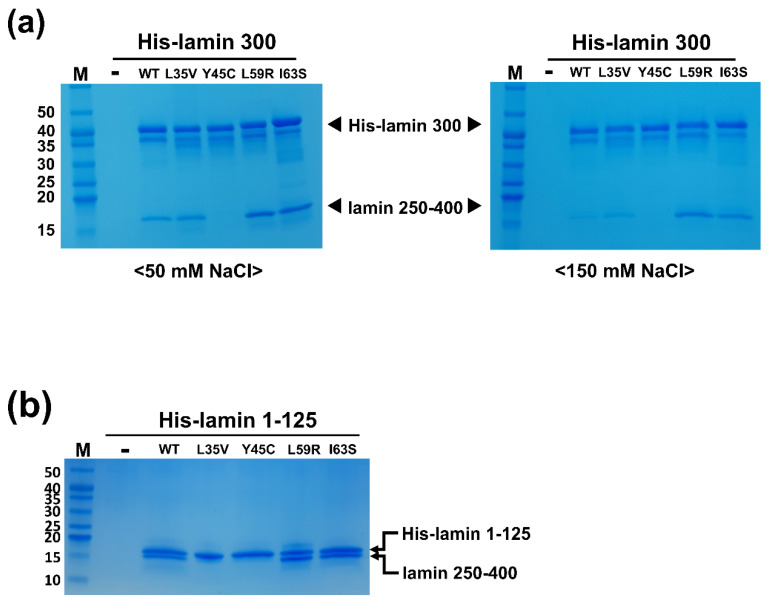
(**a**) Strength of the eA22 interaction in the four mutations related to laminopathies. The binding affinity between the lamin 250-400 (C-terminal of coil 2) and His-lamin 1-300 variants (WT, L35V, Y45C, L59R, and I63S) was analyzed by the His pull-down assays. Tag-free lamin 250-400 fragments were incubated on empty (-) or His-tagged lamin-bound (WT, L35V, Y45C, L59R, and I63S) Ni-NTA resins. The resins were pre-equilibrated and washed with 20 mM Tris-HCl (pH 8.0) buffer containing 50 mM (left) or 150 mM NaCl (right). After washing, the bound proteins were analyzed by SDS-PAGE. This is a representative result of five independent experiments. (**b**) In vitro binding assays were conducted using the immobilized wild type or mutants of lamin 1-125 fragment on Ni-NTA resin. Lamin 250-400 fragments were incubated on empty (-) or His-lamin-bound (WT, L35V, Y45C, L59R, or I63S) Ni-NTA resins. The resins were pre-equilibrated and washed with 20 mM Tris-HCl (pH 8.0) buffer containing 50 mM. This is a representative result of three independent experiments. Molecular weights (kDa) of the marker (M) are labeled on the left.

**Figure 3 cells-10-00055-f003:**
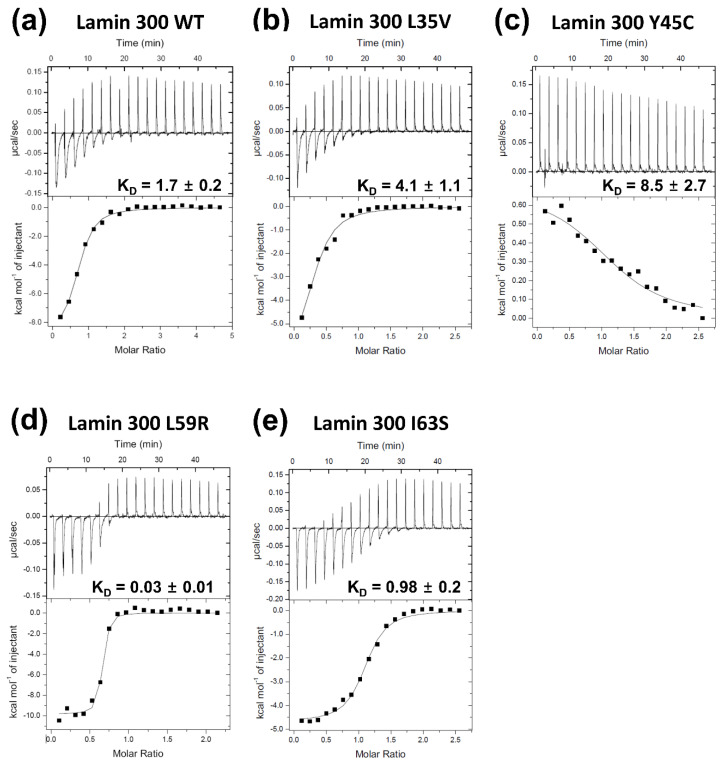
Isothermal titration calorimetry analysis for the binding affinity between the lamin 300 fragments ((**a**) WT; (**b**) L35V; (**c**) Y45C; (**d**) L59R; (**e**) I63S) and the coil 2 fragment (residues 250–400). The top panels represent the raw measured heat changes (μcal/s) as a series of peaks corresponding to a function of time resulting from the titration of each lamin 300 fragment (20 μM; 370 μL) with 19 injections of the coil 2 fragment (160 μM; 2 μL per one injection). The bottom panels show the titration isotherm resulting from the raw data in the top panels.

**Figure 4 cells-10-00055-f004:**
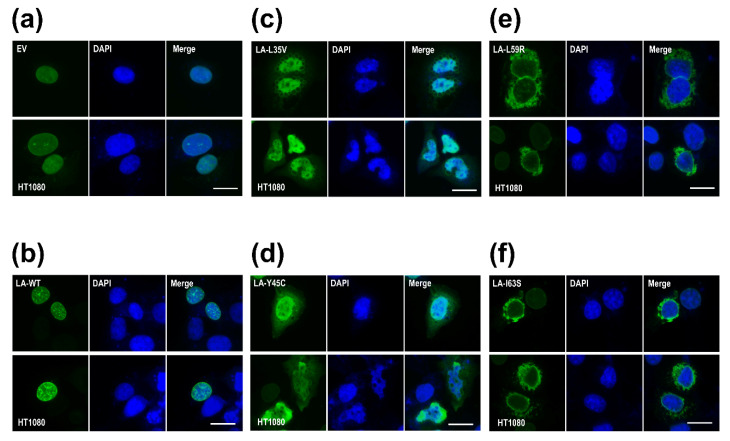
Nuclear shapes and distribution of the wild type ((**a**) empty vector and (**b**) WT) and mutants ((**c**) L35V, (**d**) Y45C, (**e**) L59R, and (**f**) I63S) of lamin A. The immunofluorescence assay visualized nuclear morphology after the transfection of wild-type or variants (L35V, Y45C, L59R, and I63S) of lamin A into HT1080 cells. For visualization of the nuclear membrane, cells were stained with an anti-lamin A/C antibody (green) and DAPI for DNA (blue). The merged images of lamin A/C and DNA are displayed on the right (merge). Scale bar: 10 μm.

**Figure 5 cells-10-00055-f005:**
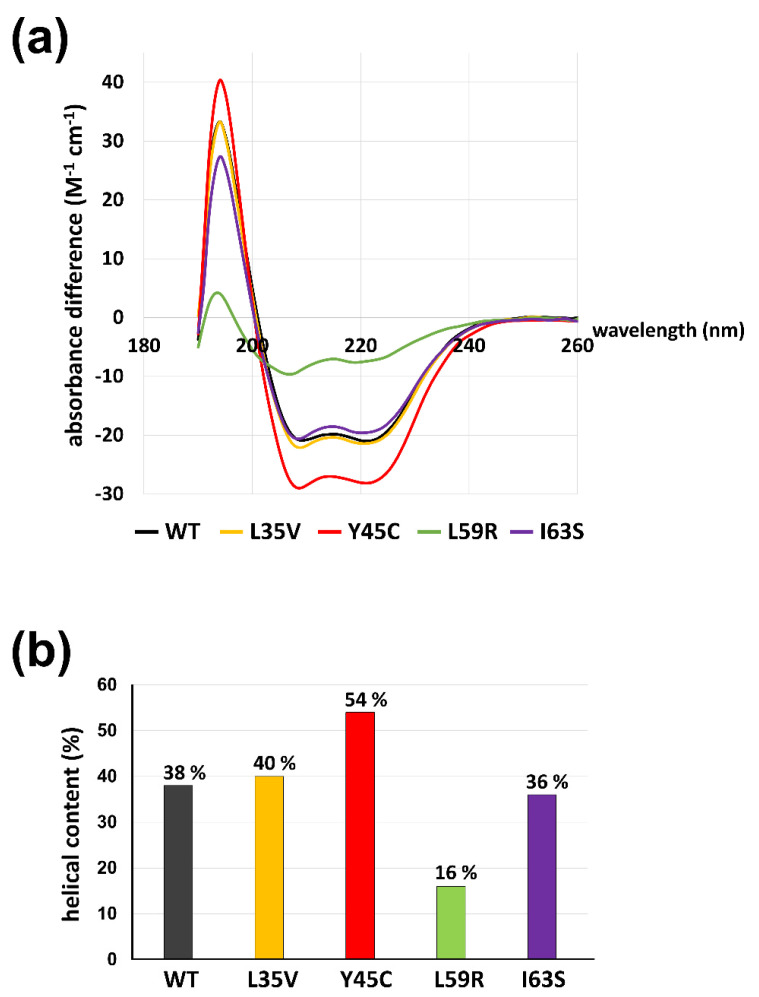
The α-helical contents of lamin 1-125 fragments (WT, L35V, Y45C, L59R, and I63S). (**a**) The circular dichroism (CD) spectra were recorded at a protein concentration of 1 mg/mL. Wild-type and variant proteins are colored differently (WT; gray, L35V; yellow, Y45C, L59R; green, I63S; violet). Estimated helical contents of lamin 1–125 fragments are presented in (**b**).

**Figure 6 cells-10-00055-f006:**
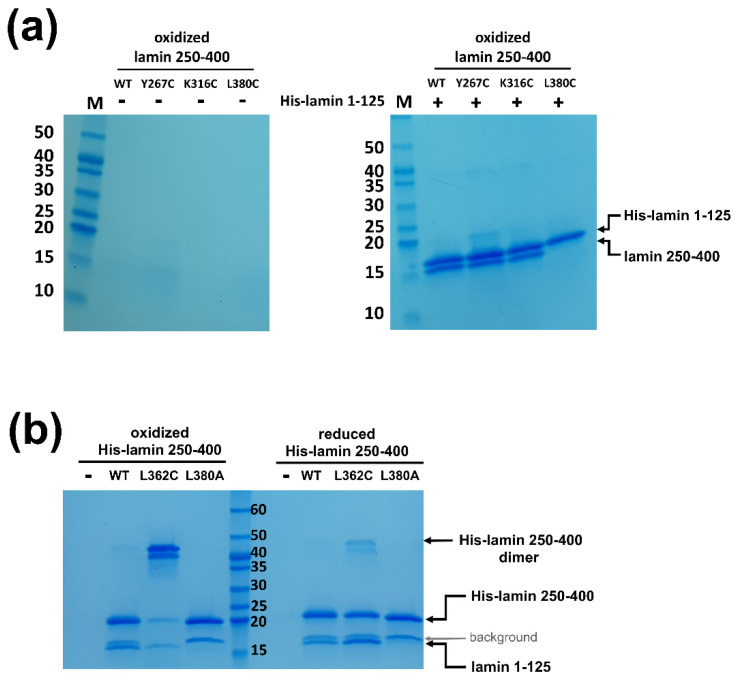
Altered eA22 interaction depending on coiled-coil dimer separation of the coil 2 fragment. (**a**) The wild type of His-tagged lamin 1-125 was immobilized on Ni-NTA resin. Lamin 250-400 fragments (WT, Y267C, K316C, and L380C) were incubated on empty (-) or His-lamin-bound (+) Ni-NTA resins under the nonreducing condition. The bound proteins were analyzed using SDS-PAGE. This is a representative result of three independent experiments. (**b**) The His-tagged lamin 250-400 fragments (WT, L362C, and L380A) were immobilized on Ni-NTA resin with the lamin 1-125 fragment. The mixed samples were incubated under the reducing or nonreducing condition. A degraded His-lamin 250-400 fragment band is shown between the molecular sizes of 15 and 20 kDa in SDS-PAGE (background, gray arrow). This is a representative result of three independent experiments. The used proteins for binding affinity are shown in Appendix A.

**Figure 7 cells-10-00055-f007:**
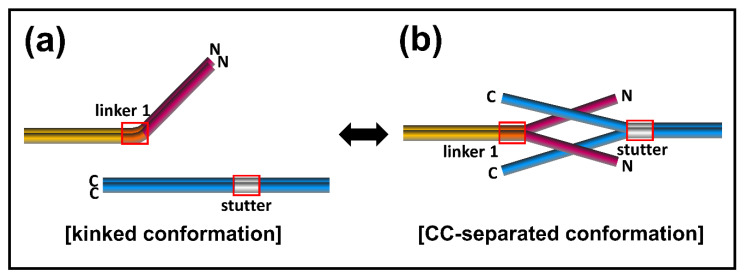
Proposed assembly model of the eA22 interaction. The kinked conformation model reflecting the lamin 300 crystal structure [18] is shown in the left panel. The coiled-coil (CC)-separated conformation model based on the IF assembly model [11,44] is shown in the right panel. Each subdomain of lamin A/C is colored differently (coil 1**a**; magenta, linker 1; orange, coil 1**b**; yellow, stutter; gray, coil 2; blue). Linker 1 and stutter are shown in a red box.

## Data Availability

Data is contained within the article or supplementary material. The data presented in this study are available in [https://doi.org/10.3390/cells10010055].

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
