# Peer review of "Separation of Coiled-Coil Structures in Lamin A/C Is Required for the Elongation of the Filament"

_cells, 2020, doi:10.3390/cells10010055_

Round 1

Reviewer 1 Report

The manuscript by Ahn et al. is a follow up study of the recent findings of the group on lamin assembly (Ahn et al., 2019; Nat Commun). Here the authors included additional mutations associated with laminopathies and investigated their influence on the eA22 interaction between coil 1a and coil 2 of two A-type lamin dimers in vitro. They mainly used bacterially expressed recombinant lamin A/C fragments representing either coil 1a or coil 2 for their studies. Based on data revealed by biochemical pull-down assays, isothermal titration calorimetry and circular dichroism analyses the authors stated that mutations either in the N-terminus of coil 1A (L35V, Y45C, L38C and L42C) or in the C-terminus of coil 2 (L362C under oxidized conditions and L380C) leading to a stronger or stable coiled-coil formation weakened the eA22 interaction, while mutations destabilizing the coiled-coil formation (L59R and I63S) had the opposite effect. In addition, the authors performed immunofluorescence experiments using HEK293 and U2OS cells ectopically expressing full length wt lamin A or various lamin A mutants (L35V, Y45C, L59R and I63S, respectively). From these experiments Ahn et al. claimed that lamin A mutants carrying eA22 interaction destabilizing mutations diffused to the cytosol, while those carrying stabilizing mutations aggregated at the nuclear periphery.

Reading the manuscript I got the impression that the authors used left-over data from their recent Nat. Commun. paper complemented with a few new results. The writing is kept very short and stringent without a lot of explanations and discussions of the presented results. I am also missing some kind of flow, so it was difficult to read the manuscript and to follow the authors intentions. Although I think the model proposed by the authors is valid and supported by the results, the presentation of the data is weak and raises many questions. Therefore I would recommend publication only after major revisions.

Major concerns:

- Figure 2: The authors state the L35V mutation decreased the eA22 interaction. However, I have the impression from the presented gels, that under low as well as high salt conditions, the mutant protein showed similar binding if not stronger compared to the wt. How often has the experiment been repeated? I think it would be good to measure the intensities of the bands revealed from several experiments, calculate the ratio between the 1-300 and 250-400 fragments and do statistics.

- Figure 3: In the text related to figure 3 the authors state, that they measured similar KD values for wt and L59R mutant. Two sentences later they write that the L53R mutant increased the eA22 interaction ~56 fold. Please clarify.

- Figures 4 and S1: The cell biological experiments are in my opinion the weakest part of the paper and if they are not improved, it does not make sense to keep them in the manuscript. The quality of the presented HEK293 images is bad (the U2O2 images are much better) and don’t allow any interpretation. Firstly, I don’t see any diffusion to the cytosol regarding the L35V and Y45C mutants. Secondly, why is emerin staining used (it is not even mentioned in the text and the staining looks strange in some untransfected cells). In addition, the mutants show different localizations in HEK293 and U2OS cells. Why have these cell lines been used? Are there any other publications investigating these mutants regarding cellular distribution? If possible I would perform confocal laser scanning microscopy with these samples.

- Figures 6, S4 and S5: Why is in figure 6a L362C missing, but shown separately in 6b together with L380A? That does not make sense to me. Dimer formation under oxidized conditions is revealed by SDS-PAGE for the L362C mutant, but not for the L38C, L42C and L380C, although the authors state that all these mutants form stable dimers under these conditions. Is this because the L362C experiments have been conducted the other way round, with the coil 2 fragments carrying a His-tag and therefore bound to the resin? If this is the case, the experiments with the other mutants (at least the mutations to C forcing S-S bridges) should be repeated that way, to see if dimers are present or not.

- Discussion: does not really discuss the results, but more the model the authors propose.

Minor comments:

- Please indicate either in the introduction and/or in the result section, which mutations are disease associated, with which diseases specifically (references) and indicate also the mutation on the DNA level.

- Figure 1a: better quality, hard to read

- line 76: period is missing.

- line 117: ITC should not be abbreviated, as it is mentioned for the first time.

- line 150: “with increased and decreased eA22 interaction” instead of “with increased and decreased the eA22 interaction”

- Figure 3: Use WT, L35V,… instead of a, b,… to label the figure.

- line 236: “affinity changes to the coil 2 region seen in ITC” instead of “affinity changes to the coil 2 region”

- line 238: “produce stronger eA22 interactions.” instead of “produce stronger the eA22 interactions.”

- line 252: Why unexpectedly? Is this not the model, that S-S bridges stabilize the dimer and therefore decrease the eA22 interaction? Or unexpectedly, because the interaction was also impaired under reduced conditions? If so, it should be shown, that there is a dimer under oxidized conditions and no under reduced ones.

Author Response

Major concerns:

Comment 1

Figure 2: The authors state the L35V mutation decreased the eA22 interaction. However, I have the impression from the presented gels, that under low as well as high salt conditions, the mutant protein showed similar binding if not stronger compared to the wt. How often has the experiment been repeated? I think it would be good to measure the intensities of the bands revealed from several experiments, calculate the ratio between the 1-300 and 250-400 fragments and do statistics.

Response: Thank you for your constructive comment. We revised the interpretation of the result in the manuscript. We repeated the binding experiment more than 5 times to compare the coil 2 binding ability between the wild type lamin fragment and the corresponding mutant fragment. If we used the lamin fragment containing residues 1-300, the binding level between the wild type and L35V were consistently very similar. However, if we used the shorter lamin fragment containing 1-125, the L35V mutant fragment apparently showed a weaker binding to the coil 2 region than the wild type, as shown in Supplementary Fig. 3b. In this revision, we incorporated the Supplementary Fig. 3b to the main Fig. 2b. Accordingly, we changed the description on the eA22 interaction between the L35V and wild type in the revised manuscript, instead of the statistical analysis. (line 162-167)

Comment 2

- Figure 3: In the text related to figure 3 the authors state, that they measured similar KD values for wt and L59R mutant. Two sentences later they write that the L53R mutant increased the eA22 interaction ~56 fold. Please clarify.

Response: We acknowledged that these sentences confused in the interpretation of the result. We revised the manuscript appropriately, as suggested by the reviewer. (line 185-186)

“The L59R mutation increased the eA22 interaction by ~56 fold according to the KD values measured in this study, consistent with the results reported in the previous study (Fig. 3a and d) [18].”

Comment 3
- Figures 4 and S1: The cell biological experiments are in my opinion the weakest part of the paper and if they are not improved, it does not make sense to keep them in the manuscript. The quality of the presented HEK293 images is bad (the U2O2 images are much better) and don’t allow any interpretation. Firstly, I don’t see any diffusion to the cytosol regarding the L35V and Y45C mutants. Secondly, why is emerin staining used (it is not even mentioned in the text and the staining looks strange in some untransfected cells). In addition, the mutants show different localizations in HEK293 and U2OS cells. Why have these cell lines been used? Are there any other publications investigating these mutants regarding cellular distribution? If possible I would perform confocal laser scanning microscopy with these samples.

Response:

We admit that the quality of the presented HEK293 images is not enough to clarify our conclusion. We think that the poor quality of the image resulted from the low expression level of endogenous lamin A. To address the reviewer’s comment, we replaced the low-quality date with the images using HT1080 cell line. We believe that the HT1080 cell line data shows the nucleus morphology related to the strength of the eA22 interactions. We removed the U2O2 cell data present in Supplemental Fig. 1, because the expression level of the Y45C mutant lamin is especially low. (See Fig. 4)

Comment 4
- Figures 6, S4 and S5: Why is in figure 6a L362C missing, but shown separately in 6b together with L380A? That does not make sense to me. Dimer formation under oxidized conditions is revealed by SDS-PAGE for the L362C mutant, but not for the L38C, L42C and L380C, although the authors state that all these mutants form stable dimers under these conditions. Is this because the L362C experiments have been conducted the other way round, with the coil 2 fragments carrying a His-tag and therefore bound to the resin? If this is the case, the experiments with the other mutants (at least the mutations to C forcing S-S bridges) should be repeated that way, to see if dimers are present or not.

Response:  To address the reviewer’s comment, we added the gel figures showing the disulfide-bond formation under oxidized conditions. (L38C and L42C in the Fig. S3a., Y267C, K316C, and L380C in the Fig. S4a)

Regarding the separate gel presentation in Fig. 6a and 6b, we admit a time gap between Fig. 6a and Fig. 6b. We tried to combine the results in a single experiment, as the reviewer commented. However, the protein samples were easily degraded during the purification. We realized that the purification of 10 proteins in a single batch was practically impossible as intact proteins. Especially the L362C mutant protein was rapidly degraded when its His-tag was cleaved. In Fig. 6b, we had to exchange the locations of the His-tag. Thus, we had to do the experiments serially in separate gels. Nonetheless, the results prove our conclusion. Fig. 6a was repeated two times, and Fig. 6b was repeated more than four times, which always produced consistent results.

Comment 5
- Discussion: does not really discuss the results, but more the model the authors propose.

Response: We added the result-summarizing paragraph in Discussion, as suggested by the reviewer. (See Discussion; line 300-307)

“In this study, we observed the altered binding affinity of the eA22 interaction by introducing the four mutations related to laminopathies into the coil 1a. Based on the structural and biochemical analysis, as a result, the eA22 interaction was weakened in the mutations, which induce more stable coiled-coil dimers in coil 1a. In contrast, the mutations that cause instability of coiled-coil (CC) dimers due to the loss of hydrophobic properties enhanced the eA22 interaction more than the wild type. Furthermore, we showed that the eA22 interaction also requires the CC-separation of the C-terminal part of coil 2. These results imply that the dynamic conformational changes in N-terminal of coil 1a and C-terminal of coil 2 are necessary for maintaining the normal function of nuclear lamin.”

Minor comments:

Comment 1
- Please indicate either in the introduction and/or in the result section, which mutations are disease associated, with which diseases specifically (references) and indicate also the mutation on the DNA level.

Response: We described the specific diseases associated with mutations in the revised manuscript. (line 149-154)

Comment 2
- Figure 1a: better quality, hard to read

Response: We re-uploaded the better quality of figure 1a, as suggested reviewer. (See Fig. 1a)

Comment 3
- line 76: period is missing.

Response: We added the period mark in the line. Thank you for the kind comment.

Comment 4
- line 117: ITC should not be abbreviated, as it is mentioned for the first time.

Response: We revised it as "Isothermal titration calorimetry (ITC)" (line 117).

Comment 5
- line 150: “with increased and decreased eA22 interaction” instead of “with increased and decreased the eA22 interaction”

Response: We replaced the sentence “with increased and decreased eA22 interaction”. (line 151)

Comment 6
- Figure 3: Use WT, L35V,… instead of a, b,… to label the figure.

Response: We appreciate these helpful comments. We added the labels on the figure 3. (See Fig. 3)

Comment 7
- line 236: “affinity changes to the coil 2 region seen in ITC” instead of “affinity changes to the coil 2 region”

Response: We revised the sentence “affinity changes to the coil 2 region seen in ITC” (line 243)

Comment 8
- line 238: “produce stronger eA22 interactions.” instead of “produce stronger the eA22 interactions.”

Response: We changed the sentence “Thus, our findings suggest that the degree of α-helix tendency might be reversely correlated to the strength of the eA22 interactions for the coiled-coil interaction would stabilize the α-helical tendency.” (line 243-245)

Comment 9
- line 252: Why unexpectedly? Is this not the model, that S-S bridges stabilize the dimer and therefore decrease the eA22 interaction? Or unexpectedly, because the interaction was also impaired under reduced conditions? If so, it should be shown, that there is a dimer under oxidized conditions and no under reduced ones.

Response: We surprised because the eA22 interaction was impaired under both oxidized and reduced conditions. As reviewer’s comments, we added SDS-PAGE data which shows dimer bands under oxidized condition and monomer bands under reduced condition. (See Fig. S4a)

Reviewer 2 Report

This short article carefully studies how lamins can interact with each other to form IF, and especially a novel kind of multimer interaction recently described. The paper is clearly written, the results are strongly supported by a cleaver experimental design and numerous data. In total, this work brings an interesting point of view on lamin polymerization and should be shared with scientific community.

Minor concerns :

  • L195: I disagree with the assertion that both cell lines show the same results. It is clear from figure S1 that L35V mutant lamins spread to the cytoplasm, but not from figure 4c... maybe this part of the results could be more precise... 
  • L206: merge images are on the right, not the left.
  • L238: unclear sentence.
  • Fig6b: ladder is missing, seems to be different from above image. Does the upper band correspond do dimers as in figure S5? That should be stated for people not reading sup material

Author Response

Reviewer 2

This short article carefully studies how lamins can interact with each other to form IF, and especially a novel kind of multimer interaction recently described. The paper is clearly written, the results are strongly supported by a cleaver experimental design and numerous data. In total, this work brings an interesting point of view on lamin polymerization and should be shared with scientific community.

Minor concerns :

Comment 1:           
L195: I disagree with the assertion that both cell lines show the same results. It is clear from figure S1 that L35V mutant lamins spread to the cytoplasm, but not from figure 4c... maybe this part of the results could be more precise... 

Response: We admit that the quality of the presented HEK293 images is not enough to clarify our conclusion. We think that the poor quality of the image resulted from the low expression level of endogenous lamin A. To address the reviewer’s comment, we replaced the low-quality date with the images using HT1080 cell line. We believe that the HT1080 cell line data shows the nucleus morphology related with the strength of the eA22 interactions.

Comment 2
L206: merge images are on the right, not the left.

Response: We correct the mistake. Thank you for your kind comment.

Comment 3
L238: unclear sentence.
Response: We changed the sentence more clearly. (line 243-245)

“Thus, our findings showed that the degree of α-helix tendency is correlated to the results for the eA22 interactions: weaker coiled-coil interaction induced by smaller α-helical content produce stronger eA22 interactions.”

Comment 4
Fig6b: ladder is missing, seems to be different from above the image. Does the upper band correspond do dimers as in figure S5? That should be stated for people not reading sup material

Response: We revised the manuscript, as suggested by the reviewer. We used the same ladder marker but different constructs with different His-tag locations. While His-lamin 1-125 WT and lamin 250-400 (WT, Y267C, K316C, L380C) were used in the upper data, Lamin 1-125 WT and His-lamin 250-400 (WT, L362C, L380A) were used in the bottom data. We added the labels indicating the disulfide-bonded dimers in the figure. (Fig. 6b)

Round 2

Reviewer 1 Report

Since the authors replied to all my comments satisfactorily and improved the manuscript, it can be accepted for publication in my opinion. However, I have only a few minor comments:

Concerning response to previous comment 1: If the authors repeated the binding experiment more than five times, I think it would be good to mention this either in the M&Ms, the results or the figure legend.

Concerning the cellular distribution of L35V and Y45C: to me it looks like that the lamin structures are in lower assembly states (dimers?) and incorporation into the nuclear lamina is impaired, therefore they localize mainly throughout the nucleoplasm but are also found to a lesser degree uniformly distributed in the cytoplasm ("diffused to the cytosol, forming localized aggregations", does not reflect the actual situation in my opinion).

Concerning the cellular distribution of L59R and I63S: they have a much higher tendency to form filaments and assemble already in the cytoplasm and therefore can't be imported into the nucleus and incorporated into the lamina: or they can't be completely dissassembled when the nucleus breaks down at mitosis and during nuclear assembly they can't be incorporated in the nucleus/nuclear lamina and stay as filamentous structures (associated with the ER?) in the cytoplasm.

However, these are just my interpretations of the transfection experiments.

Line 245: the sentence doesn't sound right.

Author Response

Comment 1

Concerning response to previous comment 1: If the authors repeated the binding experiment more than five times, I think it would be good to mention this either in the M&Ms, the results or the figure legend.

Response: We added the comments in the Figure legends (Figs. 2 and 6)

Comment 2

Concerning the cellular distribution of L35V and Y45C: to me it looks like that the lamin structures are in lower assembly states (dimers?) and incorporation into the nuclear lamina is impaired, therefore they localize mainly throughout the nucleoplasm but are also found to a lesser degree uniformly distributed in the cytoplasm ("diffused to the cytosol, forming localized aggregations", does not reflect the actual situation in my opinion).

Response: We appreciate these helpful comments. We added detailed interpretation in the revised manuscript. (line 200-204)

Comment 3

Concerning the cellular distribution of L59R and I63S: they have a much higher tendency to form filaments and assemble already in the cytoplasm and therefore can't be imported into the nucleus and incorporated into the lamina: or they can't be completely dissassembled when the nucleus breaks down at mitosis and during nuclear assembly they can't be incorporated in the nucleus/nuclear lamina and stay as filamentous structures (associated with the ER?) in the cytoplasm. However, these are just my interpretations of the transfection experiments.

Response: Thank you for the constructive comment. We added detailed interpretation in the revised manuscript. (line 204-211)

Comment 4

Line 245: the sentence doesn't sound right.

Response: We corrected the sentence as suggested by the reviewer. (line 251-253)